# Position: Bitter lesson of the ARC-AGI Challenge: Intelligence may look very different in machines and humans

**Firstname1 Lastname1** [1]   **Firstname2 Lastname2** [1]

## Abstract

The Abstraction and Reasoning Corpus (ARC) and the associated ARC-AGI challenge, serve as a benchmark to evaluate a system's ability to demonstrate core reasoning skills and abstraction. Recently, the newest, as-of-yet unreleased model by OpenAI, $o3$, passed this benchmark.

These breakthroughs have prompted discussion on whether current AI techniques are advancing genuine reasoning capabilities or engage in merely pattern-matching behaviours. The "Bitter Lesson" articulated by Rich Sutton argues that general methods with large computational resources tend to outperform specialised approaches. This insight also applies to ARC, as many winning approaches rely heavily on data augmentation and test-time training, raising philosophical and methodological questions.

We explore the tension between the goals of ARC and the methods that are applied and whether the need for new concepts and neologisms could better describe machine reasoning. Terms like "reasoning" and "understanding" may need redefinition to account for the unique characteristics of machine intelligence. This position paper argues that reasoning and intelligence may be very different in machines and humans. Recognising that intelligence is task-specific and comes in various forms may help us appreciate that animals, humans and machines have different kinds of intelligence, and help us design better benchmarks for machines.

---

[*]Equal contribution  [1]Department of XXX, University of YYY, Location, Country. Correspondence to: Firstname2 Lastname2 <first2.last2@www.uk>.

*Proceedings of the $42^{nd}$ International Conference on Machine Learning*, Vancouver, Canada. PMLR 267, 2025. Copyright 2025 by the author(s).

## 1. Introduction

The Abstraction and Reasoning Corpus (ARC) and the associated ARC-AGI challenge, proposed by François Chollet (Chollet, 2019), serve as a benchmark to evaluate a system's ability to demonstrate core reasoning skills and abstraction, qualities critical for Artificial General Intelligence (AGI). ARC focusses on reasoning tasks that require minimal prior training data and emphasises "core knowledge": a concept that reflects human-like cognitive skills. However, recent results in ARC have brought forth critical discussions about whether current AI techniques are advancing genuine reasoning capabilities or engage in merely pattern-matching behaviours.

The "Bitter Lesson" articulated by Rich Sutton (Sutton, 2019) argues that general methods with large computational resources tend to outperform specialised approaches. This insight also applies to ARC, as many winning approaches rely heavily on data augmentation and test-time training, raising philosophical and methodological questions. Does reliance on augmentation compromise the spirit of ARC, which aims to evaluate reasoning over brute-force computation?

**This paper explores the tension between the goals of ARC and the methods that are applied and whether the need for new concepts and neologisms could better describe machine reasoning. We argue that reasoning and intelligence may look very different in machines and humans. Recognising that intelligence is task-specific and may look very different in animals, humans and machines, may also help us design better benchmarks for AI algorithms**.

## 2. Defining Artificial General Intelligence (AGI)

Traditional AI benchmarks often measure task-specific skills, which can be influenced by prior knowledge and extensive training data. In contrast, ARC focusses on a system's capacity for skill acquisition and generalisation, emphasising adaptability to novel problems without prior exposure. Francois Chollet defines intelligence as "a measure of skill-acquisition efficiency over a scope of tasks, with

respect to priors, experience, and generalization difficulty" (Chollet, 2019).

ARC consists of tasks that present input-output pairs using grids where each cell can be one of ten colours (see Figure 1). The objective is to produce a pixel-perfect output grid for a given input, demonstrating the system's ability to generalise from limited examples. The tasks are designed to be solvable using basic "core knowledge priors" that humans naturally possess, such as object permanence, goal-directedness, numerical understanding, and basic geometry.

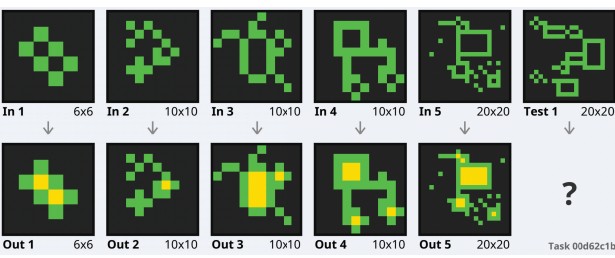

*Figure 1.* An illustration of a typical task in ARC. Each task has three training pairs shown as In (input) and Out (output) in the first three columns. The last task (denoted test) needs to be solved (shown with a question mark (?).

ARC is explicitly designed to compare artificial intelligence with human intelligence by focusing on "core knowledge priors" that humans naturally possess, even in childhood. These include:

1. Objectness: Understanding that objects persist and cannot appear or disappear without reason.

2. Goal-directedness: Recognizing that some objects are agents with intentions and goals.

3. Numbers Counting: Ability to count or sort objects using basic mathematics like addition, subtraction, and comparison.

4. Basic Geometry Topology: Understanding shapes and their transformations, such as mirroring, rotation, and translation.

ARC avoids reliance on acquired or cultural knowledge, such as language, to ensure a fair comparison between human and artificial intelligence.

## 3. Recent breakthroughs in solving ARC tasks

Recently, the newest, as-of-yet unreleased model by OpenAI, $o3$, passed a long-standing benchmark intended to evaluate AGI, the ARC-AGI challenge. The goal of the challenge was to have an AI pass at least 85% of the puzzles

in a corpus of 100 secret puzzles, using 400 (easy) training puzzles and 400 public evaluation puzzles, significantly less data than available for many other problems.

ARC was released before the advent of large-language models (LLMs) but remained very difficult for AI models until this recent announcement by OpenAI, in which $o3$ was shown to solve 87% of the semi-private ARC evaluation dataset. It is important to note that this semi-private evaluation has been shown to various APIs and therefore the evaluation set may have leaked into the training data used by OpenAI, potentially making the announcement less significant.

There remains the (theoretically) fully private dataset of 100 training problems that should have never been leaked on the internet, and it would be interesting to see the performance of $o3$ on this dataset.

We reflect on actual implications of this recent breakthrough. If ARC-AGI were indeed a benchmark perfectly measuring the general intelligence capabilities of modern artificial intelligence, and if OpenAI took precautions to ensure that none of the semi-private evaluation dataset was used in training or fine-tuning $o3$, then it could reasonably be said that OpenAI has achieved AGI.

However, it is not even clear what artificial general intelligence even means, exactly, it is not clear that ARC measures any sort of actual AGI capability, and it is not clear how much data was leaked in the creation of $o3$. We will dissect some ideas about what AGI is or could be, we will discuss the previous approaches to ARC and how general they are, and finally we will discuss ideas for future directions towards developing both general artificial intelligence benchmarks and actual general artificial intelligence systems.

Intelligence is something that is quite difficult to define, and this is something that was already understood in the design of the original ARC challenge. For example, we already have systems that are incredibly superhuman in a wide range of areas, the simplest being simple information recall—any database system can perfectly and immediately recall a vast amount of information that no human could possibly memorize.

Progressively, AI has been able to perform well even on tasks that previously seemed to require genuine intelligence, such as Chess, Go, and now ARC-AGI, but, at least up until now, these systems have not been able to generalize to the broad range of activities that humans can generalize to. As Francois Chollet recognized in the design of ARC (Chollet, 2019), while what we mean by general intelligence is vague, it likely abstractly refers to the ability to generalize to the types of tasks that humans are likely to see.

More concretely, it refers to the broad capability of humans to use their prior knowledge and experience to solve new unforeseen tasks that they may encounter. He further referred to intelligence in several different ways, for example, qualitatively categorising degrees of generalisability and considering generalisability from an algorithmic complexity perspective.

## 4. Is ARC a good benchmark anymore?

Goodhart's Law states that "When a measure becomes a target, it ceases to be a good measure." We argue that this principle is now also applicable to ARC. If optimization techniques focus on improving ARC scores through test-time augmentation, brute-force heuristics, or task-specific adaptations, the benchmark may no longer accurately assess general reasoning capabilities. This raises an important question: does ARC remain a reliable benchmark for evaluating reasoning, or has it been reduced to a target that can be gamed by specialized methods?

### 4.1. Variations on ARC tasks

We argue that the robustness of ARC-derived methods should be further tested by introducing variations that challenge their adaptability and generalisation:

1. Task variations: Modify the input-output examples while retaining the underlying concepts. For example, if the same concepts are expressed differently, can existing methods still succeed without relying on specific input-output mappings?

2. Concept Variations: Introduce tasks that explore similar concepts in novel ways, such as those inspired by ConceptARC (Mitchell et al., 2023) or string analogy problems. These new tasks would require methods to apply their reasoning capabilities to unseen scenarios while retaining their understanding of the original concepts (Mitchell, 2024b).

These variations serve to assess whether ARC-solving methods truly internalise core reasoning principles or simply exploit task-specific shortcuts. By applying such modifications, we can better understand the limits of current ARC-solving approaches and whether they are genuinely advancing the field of general intelligence.

## 5. Machines, reasoning, and the need for neologisms

A central challenge in evaluating ARC methods is defining what constitutes genuine reasoning. Is reasoning merely pattern matching at scale, or does it require an intrinsic understanding of the task?

This question reflects a broader philosophical debate:

1. Does "reasoning" in machines differ fundamentally from human reasoning?

2. Are we projecting human metaphors onto machine behaviours?

Terms like "reasoning", "understanding" and "abstraction" are laden with human-centric connotations. AI systems may operate on entirely different principles:

1. Machines might employ a spectrum of reasoning that is quantitatively and qualitatively distinct from human cognition.

2. We may be witnessing a hybrid reasoning model where machine heuristics blend with human-like abstraction.

To capture these emergent behaviours, we may require neologisms—new words that better describe machine reasoning phenomena. For example, "artificial reasoning" could be reconceptualized as "mechanical abstraction" or "synthetic inference".

Similarly, the term Artificial Intelligence itself may limit our understanding. By framing AI behaviours through familiar metaphors, we risk closing off avenues for reimagining machine intelligence.

## 6. Towards a unified theory of meaning and reasoning

The debate over reasoning mirrors the philosophical discussion of meaning: Is meaning grounded in deep understanding, or is it simply knowing how to use concepts in context?

AI systems, particularly large language models (LLMs), exhibit behaviours that challenge this dichotomy. For instance, models like OthelloGPT (Li et al., 2022) demonstrate impressive generalisation yet lack true "understanding" of the game state. This parallels the challenges in ARC: systems may solve ARC tasks using statistical shortcuts without genuinely abstracting the task concepts.

A potential resolution lies in developing a unified theory of meaning and reasoning that applies to both humans and machines. Such a theory would account for:

1. The variety of reasoning strategies across humans and machines.

2. The synergy of human-machine collaboration, where hybrid models integrate the strengths of both.

## 7. Generalisability to other datasets and other domains

Transfer from ARC to other similar benchmarks such as ConceptARC (Mitchell et al., 2023) or 1D-ARC can be used as a way to demonstrate that genuine reasoning is being performed, rather than task-specific pattern recognition.

Algorithms that play chess often solve the game through unintelligent and brute-force methods. Initially, we assumed that algorithms capable of playing chess at a superhuman level would exhibit some form of superhuman intelligence, but this assumption proved to be incorrect. These algorithms were highly specialised and skilled only within the narrow field of chess.

To truly evaluate generalisation, we propose that algorithms should be tested on other datasets, such as ConceptARC (Mitchell et al., 2023) and letter string analogies, to determine their ability to extend learned concepts to novel but related tasks.

One effective way to test the broad generalisation capabilities of AI models is by assessing them on analogous tasks within a different domain. For example, if a teacher were to examine a student's understanding of the bias-variance trade-off, the teacher would test the student by presenting a slightly different scenario that requires applying this concept. If the student is able to successfully apply the concept in this new situation, the teacher could conclude that the student has grasped the bias-variance tradeoff. Similarly, AI systems should demonstrate the ability to apply learnt principles across varied tasks to indicate true generalisation.

Similarly, if we want to test the broad generalisation capabilities of AI models, we will have to test generalisability to a similar task but in a different domain. For example, an LLM that can solve a particular type of task on ARC that has to do with an objectness prior, may be tested on a problem on the PUZZLES benchmark (Estermann et al., 2024) which has a maze. The idea being that solving a maze-like puzzle also needs some conception of objectness (i.e., while traversing a maze, you cannot pass through a wall).

We can also test the generalisation ability of these AI models by testing them on other problems such as word string analogies. For example, if an LLM can solve a permutation task on ARC, it can be tested on permutation tasks in word string analogies (Mitchell, 2024d)(Mitchell, 2019).

## 8. A call for broader agreement on intelligence

We also call for a broader inquiry and consensus on concepts such as intelligence and reasoning. In the wods of Marvin Minsky, these are "suitcase words": words that pack in a lot of concepts that are not well defined.

The term "intelligence" is deeply ambiguous, and its lack of precise definition often leads to miscommunication between artificial intelligence researchers and those from other fields (Levin & Rouleau, 2024). Researchers in AI frequently adopt functional definitions of intelligence, such as the ability to optimize solutions, process information, or achieve goals in complex environments. In contrast, researchers in fields like psychology, neuroscience, and philosophy may approach intelligence as a multifaceted phenomenon involving emotional understanding, creativity, self-awareness, or adaptive behaviour in dynamic social contexts.

For example, when AI researchers claim that a model exhibits intelligence, they often mean that the system can solve specific tasks or demonstrate performance on benchmarks such as ARC. However, a neuroscientist might argue that intelligence encompasses biological and experiential factors, such as the integration of sensory inputs and emotional states, which machines cannot replicate. This mismatch leads to debates where participants talk past each other, often disagreeing not on the empirical results but on the underlying assumptions of what constitutes intelligence.

To bridge this gap, we need to either redefine intelligence more broadly or establish precise, context-dependent definitions. A broader definition might view intelligence as the ability to navigate and adapt to complex systems, regardless of whether the agent is biological or artificial. Alternatively, a precise definition might categorise intelligence into different dimensions—such as problem solving intelligence, social intelligence, and creative intelligence—so that discussions about AI's capabilities are anchored in specific contexts.

Finally, interdisciplinary dialogue is critical. Philosophers, cognitive scientists, and ethicists should collaborate with AI researchers to develop shared frameworks and terminologies. By grounding the term "intelligence" in clear, operationalisable concepts, we can ensure that discussions about AI's potential and limitations are meaningful and less prone to misunderstanding.

## 9. Artificial general intelligence is not well defined

Artificial general intelligence (AGI) is also not well defined. Melanie Mitchell emphasises the lack of clarity and consensus around what constitutes AGI, pointing out that different researchers often use varying definitions and metrics (Mitchell, 2024a). This ambiguity makes it challenging to assess claims of AGI, particularly in light of recent advancements like OpenAI's $o3$ model.

Francois Chollet has remarked, "ARC-AGI is not an acid test for AGI ... It is a research tool designed to focus attention on the most challenging unsolved problems in AI". This

highlights that ARC-AGI is not meant to declare the arrival of AGI but to direct research toward the fundamental aspects of abstraction and reasoning.

Melanie Mitchell critiques OpenAI's definition of AGI as "all cognitive tasks that humans can do" arguing that this separates cognitive tasks from the physical world in which humans operate. For example, while ChatGPT can perform complex linguistic tasks, it cannot perform physical activities such as fixing plumbing or navigating the world as humans do. This separation reflects the familiar "brain in a vat" metaphor — the mistaken idea that intelligence can be entirely divorced from physical embodiment.

This notion harks back to Cartesian dualism where intelligence is wrongly conceived as purely cognitive (Mitchell, 2019). Modern perspectives on intelligence increasingly recognise its embodied nature, with physical interaction and sensory experience playing a critical role in developing intelligent behaviour.

The ongoing debates about AGI, fuelled by claims surrounding models such as OpenAI's o3, suggest that we may need to rethink our definitions of intelligence. It is plausible that humans, animals, and machines exhibit entirely different forms of intelligence, challenging the notion of a universal standard for evaluating intelligence.

In light of these discussions, we also call for a rebranding of the field from artificial intelligence to "actual intelligence" a suggestion echoed by Melanie Mitchell. Such a renaming could mitigate excessive anthropomorphism and better reflect the diverse manifestations of intelligence in machines and other systems. Notably, John McCarthy, one of AI's founding figures, later expressed regret about naming the field artificial intelligence. Similarly, Herbert Simon advocated for calling it "complex information processing" emphasising the computational underpinnings of the field over anthropomorphic interpretations.

## 10. There may be a spectrum of reasoning in LLMs

Reasoning and intelligence may lie on a spectrum. It is possible that reasoning in LLMs lies somewhere along this spectrum.

Reasoning might look very different in machines and humans. The word reasoning comes loaded with many assumptions: it is a metaphor and unfortunately these metaphors are used repeatedly in the field of AI (Mitchell, 2024c). We need new words (neologisms) for such emergent phenomenon in artificial intelligent systems.

## 11. Towards designing a better benchmark

Goodhart's law states that once a benchmark becomes the target of optimisation, it ceases to be a good benchmark. It is curious that the ARC challenge was solved so quickly after massive AI labs began to show interest in it, and we can assume that OpenAI probably cut some corners in the race to be the first to announce solving ARC. This observation highlights the issue with benchmarks as a whole: they are static problems. The whole point of general intelligence is that it can be used to solve problems that are completely unforeseen.

Francois Chollet attempted to make the tasks unforeseen in ARC-AGI, but the general format was quite simple,, and additionally many of the types ofproblem present in ARC-AGI could be manually reverse-engineered, to some extent defeating the idea of the challenge as testing the innate intelligence of the AI system being tested rather than the human that created it.

This lack of generality is made very clear in recent analysis (Irizar, 2024) where it is shown that simply doing an integer upscale of an ARC puzzle can make the problem intractable for an AI (in fact, the article shows some the more interesting implication that modern AI systems seem to have some maximum puzzle size that they can approach, seemingly independent of puzzle difficulty as judged by a human).

Therefore, any proper benchmark of general intelligence should itself be both as general as possible and not made available until test time. An example of a very general benchmark is SimpleBench (https://simple-bench.com/), which poses a wide variety of multiple-choice questions and which AI appears to still have difficulty approaching. However, even though not all the questions are public, they have likely been leaked through the necessity of using an API to access modern models.

In addition, although the questions are on a wide variety of topics, the general format of the test is fixed. The ideal benchmark of intelligence would likely need to be recreated every time it is used; this is expensive but necessary to ensure there are no specific priors about the benchmark incorporated in the AI system solving it. Additionally, the benchmark would need to cover as wide a variety of formats as possible: for example, one question may be a multiple-choice question akin to SimpleBench, and another question may involve frying an egg in a virtual environment.

Our current AI benchmarks are also very anthropocentric (see Figure 2). Intelligence varies depending on the task and appears in multiple forms: recognising this can enhance our understanding of the diverse intelligences present in animals, humans, and machines, and aid in developing improved benchmarks for machines.

## 12. Discussion

This paper examines the conflict between ARC's objectives and the employed methods and reflects on the whether intelligence and reasoning might ultimately look very different in machines and humans.

Some key takeaways are:

1. Goodhart's law: As ARC becomes a target for optimization, its utility as a benchmark may diminish unless new task variations are introduced.

2. Neologisms and new metaphors: We must reconsider the language we use to describe machine behaviours. Terms like "reasoning" and "understanding" may need redefinition to account for the unique characteristics of machine intelligence.

3. Unified theory: A unified approach to meaning and reasoning could help bridge the gap between human cognition and machine abstraction.

4. Reasoning and intelligence may be very different in machines and humans.

At the risk of stating the obvious, let us go over how we humans solve these visual puzzles. We solve them partly by using our visual apparatus. Specifically, light reflected from these puzzles impinges on our visual system, where part of the initial processing, such as edge detection, occurs in the eye itself. The remaining processing is carried out in the brain, which integrates this sensory input with a vast amount of prior knowledge, including concepts such as objectness, gravity, and other foundational principles derived from experience.

For machines, the processing happens in a fundamentally different way. In the case of large language models, the input is ingested as tokens in a continuous one-dimensional stream, devoid of any direct visual experience. On the other hand, if the system is a convolutional neural network, the processing may mimic certain aspects of the human visual system, such as feature detection, but it still lacks the inherent embodiment and experiential learning process of humans.

Machines are also not embodied in the physical world, meaning that they cannot acquire the same training data or learn through interaction and sensory experiences as humans. Their learning is based entirely on the data they are trained on, which is inherently limited compared to the diverse and continuous experiences humans undergo.

Hence, to expect that reasoning and intelligence in machines will mirror those of humans when processing the abstraction and reasoning corpus is a fallacy. Differences in sensory

processing, embodiment, and prior knowledge result in different pathways of reasoning and intelligence in humans and machines.

In other words, there can be many different ways to solve the same problem. *Humans and machines can solve similar problems in fundamentally different ways*.

The computer scientist Edsger Dijkstra, when asked whether computers can think like humans, famously responded with a counter-question: "Do we think submarines swim like fish?" This analogy highlights how the word "swim" is a loaded and problematic metaphor, as it imposes human-centric attributes on non-human entities. Similarly, words like "thinking," "intelligence," and "reasoning" are metaphors we frequently use, and they are what the computer scientist Marvin Minsky referred to as "suitcase words"—terms packed with a range of meanings and assumptions.

For example, while humans have always taken inspiration from birds to achieve flight, human flight looks nothing like avian flight. Airplanes achieve heavier-than-air flight using entirely different principles, such as fixed wings and engines, compared to the flapping of bird wings.

This shows that there can be multiple ways to solve the same problem, each fundamentally different from the other. For instance, the problem of heavier-than-air flight through an atmosphere—has been addressed in ways that differ profoundly between humans and birds. Similarly, the problems of intelligence and reasoning, however they are defined, may also be solved in fundamentally different ways. If broad intelligence is defined as solving problems in a complex novel environment, then machines and humans may come up with fundamentally different ways of solving these problems.

*We might need to get used to the idea that machines and humans may have very different kinds of intelligence.*

The way machines and humans approach problem-solving can be fundamentally different, yet both can lead to effective solutions. When discussing artificial intelligence, terms like thinking, intelligence, and reasoning are often applied in ways that assume machines must function like humans to be truly intelligent.

Intelligence and reasoning, however defined, need not be constrained to human-like thought processes. Machines might arrive at solutions that are equally effective, yet completely alien to human cognition.

An illustrative example comes from DeepMind's AI learning to play the Atari game BreakOut. A human playing the game would typically rely on strategies rooted in human cognitive abilities, such as tracking the ball's movement, predicting its trajectory, and using prior knowledge about similar games. However, DeepMind's AI approached the

problem differently: instead of understanding the game in the way a human would, it learned to map pixel configurations to actions.

This example highlights how machines may develop novel strategies that differ from human intuition. The AI did not "understand" BreakOut in the way a human does, yet it outperformed human players by optimising its own form of pattern recognition and action mapping. Just as airplanes do not flap their wings to fly, AI systems may not think or reason in the ways humans do—yet they may still solve problems efficiently.

*Instead of insisting that machines must mimic human cognition to be considered intelligent, we should embrace the possibility that intelligence comes in diverse forms, shaped by the constraints and capabilities of the system in which it operates.*

## 13. Alternate Views

While this paper argues that intelligence and reasoning in machines and humans are fundamentally different and need distinct definitions, alternative perspectives expand upon this view. Here we outline some alternative viewpoints:

1. The Convergence Hypothesis: Machines Will Eventually Mimic Human Intelligence.

   One view is that the differences between human and machine intelligence are primarily due to current technological limitations, rather than fundamental distinctions. Advocates of this perspective argue that, as AI systems become more advanced—particularly through improved sensory integration, embodiment, and data diversity—they will increasingly converge with human intelligence. Proponents point to ongoing advances in robotics, multi-modal AI systems, and reinforcement learning, which aim to integrate embodied learning and direct interaction with the physical world. For example, embodied AI systems, such as humanoid robots, are explicitly designed to acquire knowledge through sensory experience and interaction, mirroring human developmental processes.

   **Response**. While this perspective is compelling, it underestimates the qualitative differences in how machines and humans process information. Embodiment in machines, even when achieved, may not lead to reasoning that resembles human cognition. Unlike humans, whose reasoning is shaped by evolutionary pressures, cultural context, and physical limitations, machines operate within a fundamentally different framework constrained by their design and training data. As such, even with advanced embodiment, the processes underlying machine reasoning are likely to

diverge from those of humans. The comparison to human flight versus avian flight underscores this point: machines may achieve forms of reasoning equally valid and effective but distinct from human-like reasoning.

2. The Utility of Anthropocentric Definitions: Human-Centric Intelligence as the Gold Standard.

   Another alternative view asserts that human intelligence and reasoning should remain the gold standard for defining and evaluating machine intelligence. This position is often implicit in benchmarks like ARC, which are designed to test AI systems on tasks that are meant to reflect human cognitive abilities. Proponents argue that the ultimate goal of AI research should be to replicate and surpass human cognitive skills in a way that aligns closely with human definitions of intelligence, as this ensures practical utility in domains like healthcare, education, and governance.

   **Response.** While anthropocentric definitions have practical utility, they risk constraining our understanding of intelligence to human-specific parameters. This ignores the potential for machines to exhibit forms of intelligence that humans cannot. Such a narrow focus may lead to inefficient or misguided approaches to AI development, as it prioritizes mimicking human cognition rather than leveraging the unique strengths of machines. *By redefining intelligence to include machine-specific capabilities, we can create benchmarks and evaluation criteria that account for the distinct ways in which machines operate, fostering innovation beyond anthropocentric constraints*. ARC itself could be adapted to include tasks that explore non-human-centric reasoning, thus broadening the scope of what we consider "intelligent".

   The current situation with AI benchmarks is depicted in Figure 2, where a human is asking a seal, a penguin, a dog and a shark to climb up a tree. This ignores the fact that intelligence is task-specific: organisms have evolved over millions of years to solve tasks that are suited for the kind of environment they find themselves in. This frequently involves tradeoffs: a penguin will trade-off flight and efficient locomotion on land for swimming efficiently under water. A human (and other primates) have evolved to walk efficiently on land at the expense of not being able to fly.

These alternative views highlight the ongoing debate about the nature of intelligence and reasoning in machines. By addressing these perspectives, we reinforce the paper's central argument:

*Recognizing that intelligence is task-specific and comes in various forms and guises may help us appreciate that*

*animals, humans and machines have different kinds of intelligence, and help us design better benchmarks for machines.*

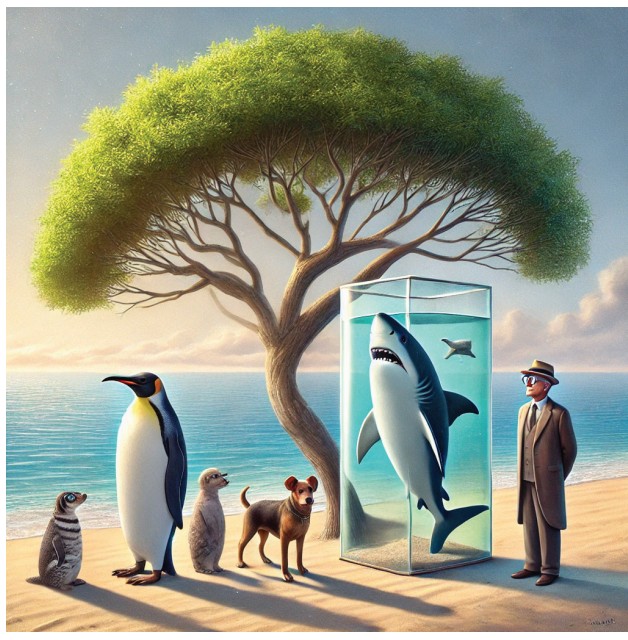

*Figure 2.* Our current benchmarks for AI are very anthropocentric. It is similar to a situation illustrated here, where a human is asking a seal, a penguin, a dog and a shark to all climb a tree. The implication is that each organism has its own strengths which have been honed through millions of years of evolution to solve certain tasks that are specific to the environment it lives in. Image generated using DALL-E.

## Impact Statement

This work highlights the unique challenges and philosophical considerations at the intersection of artificial intelligence (AI), reasoning, and abstraction. By exploring the Abstraction and Reasoning Corpus (ARC) as a benchmark, this paper emphasises the broader implications of how intelligence and reasoning are defined, measured, and understood across humans and machines.

The potential societal and ethical impact of this work lies in its re-evaluation of intelligence, urging researchers to avoid anthropocentric biases and embrace new metaphors and frameworks for describing machine intelligence. Recognising that machines and humans may solve problems in fundamentally different ways challenges existing benchmarks and calls for a more inclusive approach to assessing AI systems. This perspective could shift the focus in AI research from achieving human-like intelligence to leveraging machine intelligence in complementary ways that enhance human capabilities.

The ethical implications are significant. Misaligned benchmarks or overly anthropomorphic language could distort public and scientific understanding of AI, leading to misplaced expectations or fears. By addressing these issues, this paper advocates for a more transparent and accurate representation of machine reasoning and intelligence. This will foster trust in AI systems.

By advancing a nuanced understanding of reasoning and intelligence, this work aims to inform AI research, policy, and public discourse, encouraging ethical development of AI systems that respect the diversity of intelligence across humans, animals, and machines.

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
