# OpenReview forum: "Position: Bitter lesson of the ARC-AGI Challenge: Intelligence may look very different in machines and humans"
_ICML.cc/2025/Position_Paper_Track — Submitted to ICML 2025 Position Paper Track_

### Official Review · Reviewer_XA4W · 2025-03-05

**Significance:** 2
**Argument Clarity:** 1
**Rating:** 2
**Confidence:** 4

**Questions:**

n/a

**Discussion Potential:**

2

**Paper Summary:**

* The paper discusses how to follow up after the overcoming of ARC-AGI benchmark (which has been generally resistant) by openai o3.
* The paper questions whether reasoning and intelligence are fundamentally different in humans vs machines - and if we need neologisms for machine reasoning or not.
* The paper proposes potential follow-up benchmarks - like concept-ARC, or modified input-output while retaining underlying concepts.
* The paper calls for a broader agreement on the definition of intelligence and AGI.
* The paper cautions that our benchmarks can be too anthropocentric, and moreover, that intelligence is task-specific.

## Updates after rebuttal:

No rebuttal.

**Position:**

Yes

**Position In Title:**

Yes

**Related Work:**

2

**Strengths And Weaknesses:**

**Strengths:**
* It tackles some important points (e.g. about what to follow up after ARC-AGI)
* Many of the topics consider are discussion-friendly topics - such as how to construe machine reasoning, how to define AGI/intelligence, or whether intelligence should be considered "task-specific"

**Weakness:**
1. Most claims are not well supported.
1. The paper goes to and fro between what feels like disparate points without a cohesive overarching structure.
1. The paper tries to tackle different talking points but not many are well fleshed out. Many of the things said are common knowledge in AI community (like that AGI is not "well-defined"), and does not offer any significantly new framework or insight (for instance, what would be the concrete next steps, beyond just the overarching generic goal (such as that we should agree on a definition with interdisciplinary collaboration) or offer new critiques on what exactly are the issues with earlier proposed definitions like Chollet's or others. Thus, even if the topics themselves are discussion-friendly, too little is done in this paper to contribute to the discussions.
1. A good percentage of the paper is citing and echoing papers by Mitchell et al.  - limiting the value of the paper as a unique contribution over and beyond what's already said and presented by other researches.

**In-depth critique:**

### Sec 5

> Is reasoning merely
pattern matching at scale, or does it require an intrinsic
understanding of the task?

What exactly is "intrinsic understanding"? And what is the argument or justification that is a real dichotomy?

> Machines might employ a spectrum of reasoning that
is quantitatively and qualitatively distinct from human
cognition.

The paper, arguably, lacks clear takeaway point from this, or clarity on even how to exactly construe this.

First of all, it is common knowledge machine reasoning, if we are considering implementation details, would be different than human reasoning. For example, humans rely with biological brains, with continuous spike trains, neurotransmitters etc. whereas current machines generally rely on silicone-based architectures, transistors, and such. No one relevant is under the misconception that the implementation details under the hood is same between human and machines.

Moreover, it's also would not be a very surprising point that there are some differences in how LLMs "reason" and how humans do - given the drastic differences in architectures, the pre-training context, tokenization schemes and so on. It would be more surprising if there were no functional differences despite all that.

If that's all that is being said or suggested here - it feels like just "common knowledge" is being presented as some non-trivial point to take into consideration.

When someone would say that the principles of reasoning are similar in some current LLM (or can be similar in some potentially an ideal future AI) and humans are same/similar, they are mostly like making a functionalist point [1] - that is arguing that functional form of reasoning at the right level of abstraction [2] instantiated by the machine and the human would be similar.

So one interpretation of the quote could be that the authors are trying to argue against functionalism. This make this point more interesting.  But the authors  neither provide any specific argument against functionalism, but instead even seems to side with functionalists, by adopting functionalist metaphors later down the paper (such as how despite both birds and airplanes flying in different manner, both can fly -- i.e. implementing the same "function" at sufficient level of abstraction (after we ignore variances in implementation details sufficiently)).

All that makes it even more confusing what to take away from this. And the authors seem to neither support functionalism with confidence, nor provide any arguments for it (just putting it out as a possibility).

> To capture these emergent behaviours, we may require neologisms—new words that better describe machine reasoning
phenomena. For example, “artificial reasoning” could be
reconceptualized as “mechanical abstraction” or “synthetic
inference”.

Next, the authors argue that we may want to use neologisms to distinguish machine reasoning from human reasoning. However, there is not much provided in support why that's needed to be done, and why can't there be any underlying commonality of principles to which reasoning can refer to. The authors say that the standard term "reasoning" may have too much of a human connotation, but that's not very obvious. Moreover, even if there is some connotation, metaphilosophically it's not obvious that we have to constantly keep on creating neologisms or create new concepts just for some undesired connotations that some terms accumulate.

Very limited motivation, overall, are provided why we want to create the neologisms.

Afterwards, just in the next section, the authors themselves seem to forget the previous point and argue we should be making a more generalized notion of reasoning that accounts for both machine and human reasoning. Do we use another neologism for that? Like "general reasoning"? Again, it's not clear why we can't just conceptually "explicate" the existing term "reasoning" to serve that purpose instead of proliferating multiple neologisms. Most natural pre-scientific terms can be fuzzy, and as terms evolve based on scientific progress, certain past connotations (and sometimes even denotative aspects) naturally falls off. Such points are discussed by Carnap and later philosophers in trying to provide a systematic framework for constructing or explicating concepts - with holistic considerations - like elegance, fertility, similarity with existing usage etc. [3, 4].

Moreover, the whole discussion about a more broader topic on this line i.e. whether we want to move away for attributing intentional states to machines and current AI (or not) is already presented in a previous paper [5].

### Sec 6

> The debate over reasoning mirrors the philosophical discussion of meaning: Is meaning grounded in deep understanding, or is it simply knowing how to use concepts in
context?

Citation? Which philosophical discussion specifically? This seems too vague to map into any actual philosophical debate that exist. And pretty sure, not many philosopher would take this dichotomy.

> AI systems, particularly large language models (LLMs), exhibit behaviours that challenge this dichotomy. For instance,
models like OthelloGPT (Li et al., 2022) demonstrate impressive generalisation yet lack true “understanding” of the
game state. This parallels the challenges in ARC: systems
may solve ARC tasks using statistical shortcuts without
genuinely abstracting the task concepts.

Needs elaboration. How does this challenge the dichotomy?
Why, despite "impressive generalization", we are supposed to believe it lacks "true understanding" of the game state. These kind of claims does not seem to exist in the cited paper on my first glance.

### Sec 7

> 1D-ARC

Needs citation.

### Sec 8

> In the words of Marvin
Minsky, these are “suitcase words”: words that pack in a lot
of concepts that are not well defined.

Needs citation.

> Finally, interdisciplinary dialogue is critical. Philosophers,
cognitive scientists, and ethicists should collaborate with
AI researchers to develop shared frameworks and terminologies. By grounding the term “intelligence” in clear,
operationalisable concepts, we can ensure that discussions
about AI’s potential and limitations are meaningful and less
prone to misunderstanding.

This section is okay but feels a bit generic. Most of these are just common points that get repeated. Interdisciplinary collaboration is ok, but also a very generic suggestion - that anyone would think of. The section fails to provide any new insight about how to start developing a more general notion of intelligence. Also lacks any discussion about existing definitions and attempts, - like Chollet's effort (in the paper you already cited) or Weinbaum et al's [6]. Specifically there is a lack of discussion on what is exactly lacking in them, and what needs to be improved, why cannot any of them already be used?

This can be a whole topic in itself and you may not get space to fit all that discussion, but that just goes back to the weakness I cited. The paper rushes through several topics, none particularly discussed in depth.

Moreover, some of this topics like defining intelligence, or AGI seems a bit disconnected, from say creating a new dataset, or creating neologisms.

### Sec 9

> Melanie Mitchell critiques OpenAI’s definition of AGI as
“all cognitive tasks that humans can do” arguing that this
separates cognitive tasks from the physical world in which
humans operate. For example, while ChatGPT can perform
complex linguistic tasks, it cannot perform physical activities such as fixing plumbing or navigating the world as
humans do. This separation reflects the familiar “brain in a
vat” metaphor — the mistaken idea that intelligence can be
entirely divorced from physical embodiment.
This notion harks back to Cartesian dualism where intelligence is wrongly conceived as purely cognitive (Mitchell,
2019). Modern perspectives on intelligence increasingly
recognise its embodied nature, with physical interaction
and sensory experience playing a critical role in developing
intelligent behaviour.

For the sake of self-containment, these sections require more explicit discussions about what the critique in Mitchell et al's works are.
For instance:
* Why do concrete-actuator-involved tasks need to be considered for evaluation of intelligence? Why can't only virtual actions be enough?
* While because of implementation details, human intelligence and functions can depend on bodily states (hunger, stress, pressure etc.), why can't evaluation of intelligence be focused on cognitive actions more causally proximate to the core cognitive processes at the moment? Why would it be believed that intelligence is not just in practice but in principle necessarily entangled with the body? (in fact the authors own suggestions of thinking that intelligence can come in various shapes and forms seem to be a caution against believing in such constraints)
* Why is brain in a vat a myth? Why can't it or something like it work out if all the bodily feedback is provided just by a mechanical system instead of the organic body?
* Modern perspectives part needs to be cited. For instance, all the works related to embodied cognition and such. However, note they are not completely uncontroversial and settled positions either.
*  "embodied nature, with physical interaction
and sensory experience" - needs more clarity. Programs are still instantiated in some physical system, LLMs can get "sensory" input as tokens, and can physically interact with us via out generations and taking input from us.  Based on non-arbitrary rules would one qualify as "embodied" and another "not"?

### sec 10

Seems redundant. What is said in section 10 is already said in earlier sections. Also too small of a section to be a separate section.

### sec 11

Seems to have some redundancies with section 7. Potentially can be combined into a single section.

> Our current AI benchmarks are also very anthropocentric
(see Figure 2). Intelligence varies depending on the task and
appears in multiple forms: recognising this can enhance our
understanding of the diverse intelligences present in animals,
humans, and machines, and aid in developing improved
benchmarks for machines.


Again this is a bit vague and does not translate to a clear takeway.
What exactly is considered as anthropocentric? What about theorem proving, formal logical inference, protien folding, genomics, and all of kind of synthetic tasks to test very specific generalization capacities such as flip-flop, string reversing etc.? Are all of them anthropocentric despite being much different from what lay human generally engage in? What kind of tasks would qualify as non-anthropocentric and how can they be designed and utilized?

Just a vague remark about not testing a fish by its ability to climb a tree does not do much good.

Moreover, if the intention is to go towards AGI, we potentially want it to be capable of learning the needed prior to do good in both in anthropocentric and non-anthropocentric tasks.

### Sec 13

> The current situation with AI benchmarks is depicted
in Figure 2, where a human is asking a seal, a penguin,
a dog and a shark to climb up a tree. This ignores the
fact **that intelligence is task-specific**: organisms have
evolved over millions of years to solve tasks that are
suited for the kind of environment they find themselves
in. This frequently involves tradeoffs: a penguin will
trade-off flight and efficient locomotion on land for
swimming efficiently under water. A human (and other
primates) have evolved to walk efficiently on land at
the expense of not being able to fly

Throughout the paper, there is the claim that intelligence is task-specific. But it's generally just mentioned in the passing.
And sometimes the paper also talks about generalization and avoiding "task-specific shortcuts" which seems to be tension with the notion of task-specificity of intelligence. Either way, it's again not elaborated that well what is this supposed to mean. Is the claim here that a general notion of intelligence is in-principle ill-formed (not just ill-defined)?

If so, again, this is an interesting possibility but this requires more defense with multiple supporting points. Just pointing to penguins having task-specific intelligence (even if that's so) doesn't go much far in justifying that intelligence in-principle cannot be universal or general -- but only be task-specific or be applicable to only a short range of tasks due to trade-offs (improving one set of tasks decreases skill in other set of tasks).

Humans already seems to hint towards a degree of task-agnostic generality, and humans may not be the endpoint in maximum generalization. There's also arguments from g-factor, and Marcus' Hutter's AIXI [7] and so on - none of which are considered.




**Additional Feedback**

* Around sec 3, you can also bring up the point about whether o3 and such should be valid candidates - as in should they count as having just minimal "core knowledge priors" given extensive priors. Following Chollet's definition the original intention was to see how much broad task performance can be achieved with limited priors. But that point is somewhat betrayed if you a model with extensive pre-training. Although, in those terms it's also hard to judge what to consider as a good baselines, since once could argue humans also go through extensive "pre-training" (potentially via predictive processing on constant sensual data) before getting some of the core "priors" (like object permanence). All these also make it difficult to understand what precisely would be the takeaway point from success/failure of models in such benchmarks.

* The author's suggestions for follow up benchmarks for AGI-ARC are good. However, one of the main suggestions is just representing an already proposed benchmark by Mitchell et al (ConceptARC). Moreover, AGI-ARC-2 is already underway. So, limited impact poential from the suggestions.

* Would be good to add more citations. At least cite the resource about o3 beating ARC-AGI.

[1] https://plato.stanford.edu/entries/functionalism/

[2] The Method of Levels of Abstraction - Luciano Floridi, Mind and Machines 2008

[3] https://iep.utm.edu/explicat/

[4] Carnap, R., 1950, Logical Foundations of Probability, Chicago: The University of Chicago Press.

[5] Talking about Large Language Models - Murray Shanahan, Communications of the ACM 2024

[6] Weinbaum (Weaver), D., & Veitas, V. (2016). Open ended intelligence: the individuation of intelligent agents. Journal of Experimental & Theoretical Artificial Intelligence, 29(2), 371–396. https://doi.org/10.1080/0952813X.2016.1185748

[7] An Introduction to Universal Artificial Intelligence - Hutter et al. 2024

**Support:**

1

---

### Official Review · Reviewer_mQ8s · 2025-03-08

**Significance:** 2
**Argument Clarity:** 1
**Rating:** 1
**Confidence:** 4

**Questions:**

Please see Weaknesses.

**Discussion Potential:**

2

**Paper Summary:**

This paper argues that despite recent advances in the ARC-AGI benchmark, we still cannot conclude that artificial general intelligence has been achieved. It then presents several perspectives:

1. ARC-AGI alone is not a sufficient benchmark; variations should be used (Section 4).
2. Machines and humans may reason differently, which could require distinct definitions of reasoning (Sections 5, 6).
3. AI models should be tested on additional datasets and domains beyond ARC (Section 7).
4. AGI itself is not well-defined (Section 9).

**Position:**

Yes

**Position In Title:**

Yes

**Related Work:**

1

**Strengths And Weaknesses:**

Strengths:

1. The paper highlights important issues, such as the limitations of existing benchmarks and the lack of a clear definition for AGI.
2. The authors present several interesting viewpoints.

Weaknesses:

1. The argument that ARC-AGI is an insufficient benchmark for AGI is well known (see discussions in [[1](https://www.lesswrong.com/posts/KHCyituifsHFbZoAC/arc-agi-is-a-genuine-agi-test-but-o3-cheated), [2](https://autogpt.net/arc-agi-test-nears-resolution-but-experts-question-its-validity-in-agi-quest/)], among others).
2. The paper presents perspectives but lacks strong arguments and supporting evidence. For example, the claim that "AI models and humans reason differently" is plausible but not substantiated.
3. With 13 sections, the paper lacks an outline explaining their necessity and how they connect.

**Support:**

2

---

### Official Review · Reviewer_UpoN · 2025-03-13

**Significance:** 2
**Argument Clarity:** 3
**Rating:** 2
**Confidence:** 4

**Questions:**

None

**Discussion Potential:**

2

**Paper Summary:**

This paper discusses the ARC-AGI benchmark in the context of OpenAI o3's recent breakthrough results on it. It argues that terms like "reasoning" and "understanding" may need to be redefined to account for the unique characteristics of machine intelligence. First it summarizes the ARC-AGI benchmark and recent o3 results on it. Then it touches upon a number of different topics about intelligence and definitions for AI. The paper:
- Says Intelligence is already difficult to define
- Suggests making various variations to ARC "to understand the limits of current ARC-solving approaches"
- Reflects on the difficulty of defining reasoning in humans and machines, and suggests we might need new definitions for understanding these phenomena
- Talks about evaluating models on other datasets.
- Calls for more precise definitions of intelligence.
- Calls for more precise definitions of AGI
- Discusses how to design a new AGI benchmark by having as many different domains as possible and not making the test available until test time.

**Position:**

Yes

**Position In Title:**

Yes

**Related Work:**

2

**Strengths And Weaknesses:**

The position of the paper has the potential to be interesting, inspire discussion, and very interesting to the community. However, I don't think the evidence and justification in the paper is able to live up to the position and provide very interesting content.

The argument for variations to ARC do not seem to understand that ARC already is implicitly testing with novel tasks and patterns not seen during training. So by performing well on ARC's evaluation set, o3 shows that it is able to generalize in the ways the author proposes as variations to ARC tasks (section 4.1)

The paper does not really justify why human and machine reasoning might differ, other than the performance of o3 on ARC.

The paper suggests evaluating models on other datasets to measure generalization to novelty better. But, I think ARC already does this to some extent. In fact, I don't think the paper has a good understanding of AI's current abilities. For example, it suggests testing a model on permutations in word string analogies. But we already know reasoning models can solve these analogies. Furthermore, it would be extremely easy for the authors to test this themselves with existing models.

The paper calls for more precise definitions of intelligence. But does not offer any suggested definitions precisely. It mentions generalization and broad abilities, and adapting to novel tasks not seen during training, but these are already commonly held views, and the paper does not offer much novel discussion on the ideas.

The paper discusses how to design a new AGI benchmark by having as many different domains as possible and not making the test available until test time. Again, I think ARC-AGI is already aiming for these things. The paper would do a better job if it clarified how ARG-AGI is not satisfactory.

**Support:**

1

---

### Official Review · Reviewer_rLuH · 2025-03-20

**Significance:** 3
**Argument Clarity:** 2
**Rating:** 4
**Confidence:** 4

**Questions:**

Can you make sections 12 and 13 more scientific? E.g. for the views you propose, surely some of these (perhaps in different form) have already been mentioned in the literature?

**Discussion Potential:**

4

**Paper Summary:**

The paper reviews the ARC-AGI challenge (and similar ones), as well as notions of intelligence (in particular AGI). Their position is that the definitive intelligence needs to be reviewed

**Position:**

Yes

**Position In Title:**

Yes

**Related Work:**

3

**Strengths And Weaknesses:**

Pros:
- while the position is not a surprising one, and is "folklore" in ML, the paper nonetheless provides a compellingly written account of why the notion of intelligence does not support current ML research well.
The authors quote well-known figures from the past, such as Herbert Simon, who advocated calling AI "complex information processing" rather than "AI".
- The paper also references Goodhart's law, which was an original connection to make with respect to ARC-AGI.

Cons:
- the authors acknowledge that o3 might have cheated on ARC-AGI. I think a better reference to this claim would have helped. For example https://www.lesswrong.com/posts/KHCyituifsHFbZoAC/arc-agi-is-a-genuine-agi-test-but-o3-cheated gives more context.
- some interesting ideas state, but unexplored
- typos: "in the wods of Marvin Minsky" -> words; "the article shows some the more interesting application" -> word "some" is superfluous
- references missing in the discussion section and overly vague (e.g. in would support the text when claims are made how humans solve these; and to ground the vague text about how light reflected from the puzzles is processed in the brain instead with more scientific information about how the brain works). The discussion section and alternative views in general seems of lower quality than the rest of the paper, making me wonder if a different author wrote it?

**Support:**

3

---

### Decision · Program_Chairs · 2025-04-27

**Decision:**

Reject

**Comment:**

Three out of the four reviews recommended rejection (two weak rejects, one reject) and presented several detailed comments and concerns. However, the authors did not submit any rebuttal, so they did not use the opportunity to respond to and clarify these comments and concerns. Consequently, I think the only possible recommendation is to reject this submission.